# A catalog of small proteins from the global microbiome

Yiqian Duan[1], Célio Dias Santos-Júnior [1,2], Thomas Sebastian Schmidt [3,12], Anthony Fullam[3], Breno L. S. de Almeida [1], Chengkai Zhu[1], Michael Kuhn [3], Xing-Ming Zhao [1,4,5,6,7] ✉, Peer Bork [3,8,9] & Luis Pedro Coelho [1,10,11] ✉

Small open reading frames (smORFs) shorter than 100 codons are widespread and perform essential roles in microorganisms, where they encode proteins active in several cell functions, including signal pathways, stress response, and antibacterial activities. However, the ecology, distribution and role of small proteins in the global microbiome remain unknown. Here, we construct a global microbial smORFs catalog (GMSC) derived from 63,410 publicly available metagenomes across 75 distinct habitats and 87,920 high-quality isolate genomes. GMSC contains 965 million non-redundant smORFs with comprehensive annotations. We find that archaea harbor more smORFs proportionally than bacteria. We moreover provide a tool called GMSC-mapper to identify and annotate small proteins from microbial (meta)genomes. Overall, this publicly-available resource demonstrates the immense and underexplored diversity of small proteins.

Small open reading frames (smORFs) are found in all three domains of life, estimated as 5–10% of annotated genes[1–3]. Small proteins encoded by smORFs have been reported to perform key functions in microbial cells[4–8] and have been found involved in transcription to regulate gene expression[9], to stabilize large protein complexes[10], in signaling transduction pathways[11], regulation of transporters[12], sporulation[13,14], photosynthesis[15], and response to environmental cues[16]. In addition, small proteins can also perform antibacterial activities[17] or compose toxin/antitoxin (TA) systems[18,19].

However, small proteins have been neglected in (meta)genomics-based global studies of the microbiome[20,21] due to the difficulty in reliably identifying smORFs using genomic information alone[22,23]. Advances in Ribo-Seq[24] and proteogenomics methods[25,26] combined with comparative genomics methods[27,28] have enabled the discovery of

an increasing number of small proteins in various microorganisms[29–32]. For example, a recent systematic study revealed 4539 novel conserved small protein families of the human microbiome[33], 30% of which are predicted to encode transmembrane or secreted proteins. However, most of the studies focusing on smORFs approach isolated microorganisms and specific environments. The functional and ecological understanding of microbial smORFs at a global scale across different habitats is still very limited.

Here, we use the principle that repeated independent observations of the same small protein (or minor variations thereof) minimize the likelihood of false positive smORF predictions and construct a global microbial smORFs catalog (GMSC) derived from 63,410 assembled metagenomes from the SPIRE database[21] and 87,920 isolate genomes from the ProGenomes2 database[34]. In the catalog, we provide

[1]Institute of Science and Technology for Brain-Inspired Intelligence, Fudan University, Shanghai, China. [2]Laboratory of Microbial Processes & Biodiversity - LMPB; Department of Hydrobiology, Universidade Federal de São Carlos – UFSCar, São Carlos, São Paulo, Brazil. [3]Structural and Computational Biology Unit, European Molecular Biology Laboratory, Heidelberg, Germany. [4]Department of Neurology, Zhongshan Hospital, Fudan University, Shanghai, China. [5]Lingang Laboratory, Shanghai 200031, China. [6]State Key Laboratory of Medical Neurobiology, Institutes of Brain Science, Fudan University, Shanghai, China. [7]MOE Key Laboratory of Computational Neuroscience and Brain-Inspired Intelligence, and MOE Frontiers Center for Brain Science, Fudan University, Shanghai, China. [8]Max Delbrück Centre for Molecular Medicine, Berlin, Germany. [9]Department of Bioinformatics, Biocenter, University of Würzburg, Würzburg, Germany. [10]Centre for Microbiome Research, School of Biomedical Sciences, Queensland University of Technology, Translational Research Institute, Woolloongabba, QLD, Australia. [11]Centre for Data Science, Queensland University of Technology, Brisbane, QLD, Australia. [12]Present address: APC Microbiome and School of Medicine, University College Cork, Cork, Ireland. ✉e-mail: xmzhao@fudan.edu.cn; luispedro@big-data-biology.org

comprehensive annotation containing taxonomy classification, habitat assignment, quality assessment, conserved domain (CD) annotation, and predicted cellular localization. In addition, the catalog can be used as a reference to annotate (meta)genomes as the presence of homologs reduces the probability that false positives are reported. To facilitate this, we developed a tool named GMSC-mapper, which additionally provides users with information about the distribution of any matching smORFs across taxonomy, habitats, and geography. Thus, our catalog and associated tools can be used to study the presence, prevalence, distribution, and potential ecological roles of smORFs on a global scale, and provide new insights into how these molecules work within microorganisms.

## Results

### The global microbial smORFs catalog comprises 965 million smORFs

The global microbial smORFs catalog (GMSC) was derived from 63,410 publicly available assembled metagenomes spanning multiple habitats worldwide from the SPIRE database[21] and 87,920 high-quality isolate microbial genomes from the ProGenomes2 database[34] (Fig. 1a, Supplementary Data 1). From the assembled contigs, we used the modified version of Prodigal[35] in Macrel[36] to predict open reading frames (ORFs) with a minimum length of 30 nucleotides (see Methods). The ORFs encoding small proteins (here defined as those up to 100 amino acids) were considered small ORFs (smORFs).

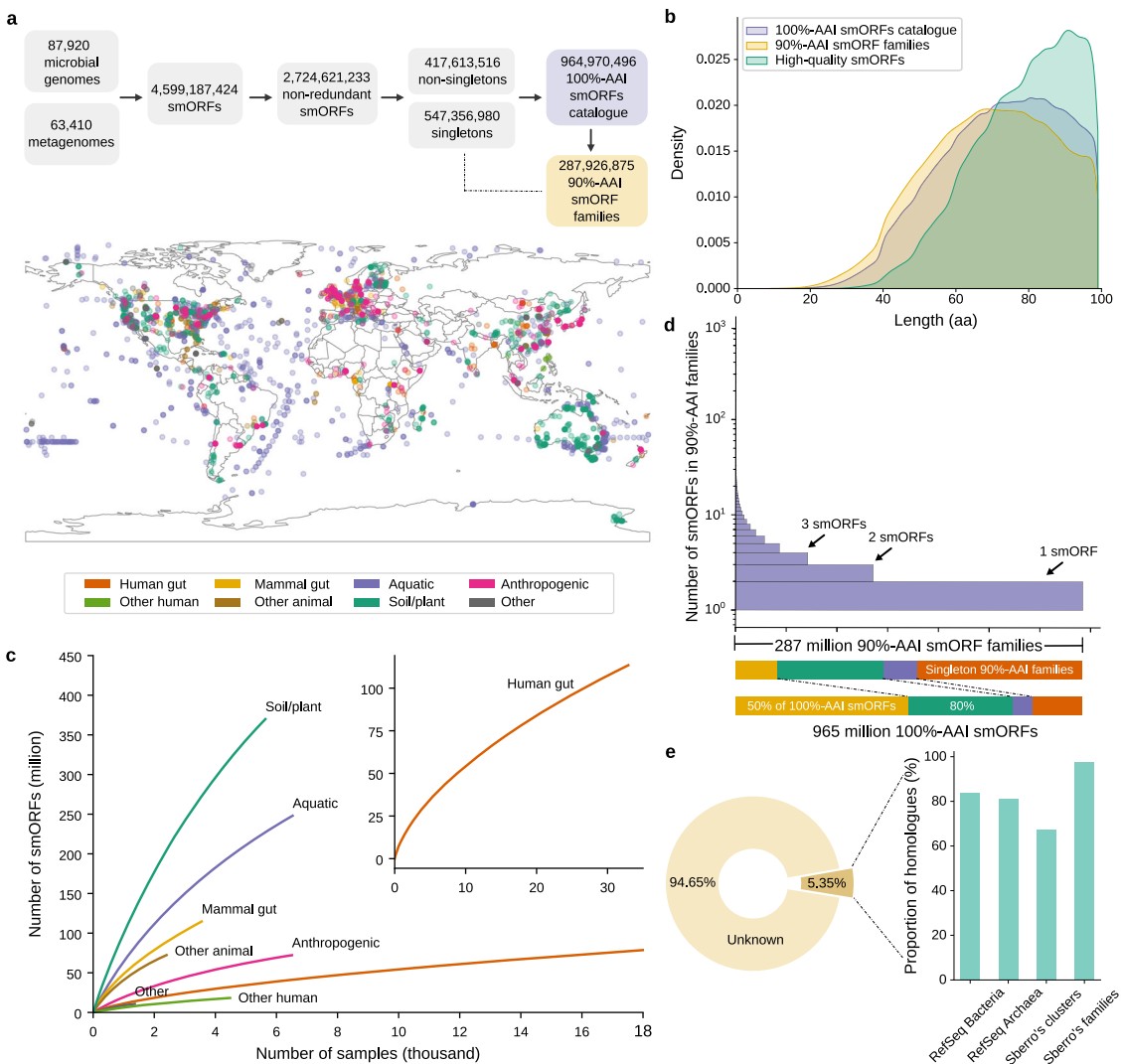

**Fig. 1 | Global Microbial smORFs Catalog (GMSC). a** ORFs (open reading frames) were predicted from contigs from 63,410 assembled metagenomes from the SPIRE database and 87,920 microbial genomes from the ProGenomes2 database. The ORFs with at most 300 bps were considered smORFs. In total, 4,599,187,424 smORFs were predicted, of which 99.25% originated in metagenomes and 0.75% originated in microbial genomes. The number of smORFs was reduced to 2,724,621,233 by removing redundancy at 100% amino-acid identity (AAI) and 100% coverage. We further clustered the non-redundant smORFs into 287,926,875 clusters at a 90% amino-acid identity (AAI) cutoff (Methods). **b** Small proteins encoded by smORFs range in length from 9 to 99 amino acids. Sequences that pass all in silico quality tests and contain at least one piece of experimental evidence are considered high-quality predictions (Methods). **c** Shown are gene accumulation curves per habitat, showing how sampling affects the discovery of smORFs (see also

Supplementary Fig. 2a). **d** The largest 90%-AAI smORF family contains 4577 sequences. The size of 90%-AAI smORF families exhibits a long tail distribution, and 47.5% of families consist of only one sequence, accounting for fewer than 15% of the total GMSC smORFs. A small fraction of large families account for the majority of GMSC smORFs (12.2% of families contain 50% of smORFs). **e** Only 5.35% of smORFs in the GMSC have a homologous sequence in another sequence catalog (Methods). On the other hand, more than 80% of bacterial and archaeal small proteins from the RefSeq database have a homolog in our catalog. Although only 67.3% of the 444,054 small protein clusters from the Sberro human microbiome dataset are homologous to a protein in our catalog, most of their clusters without homologous sequences only contain one sequence. Among the 4539 conserved small protein families from the Sberro human microbiome dataset, 97.4% of them are homologous to our catalog.

In total, after collapsing smORFs coding for identical amino acid sequences, we obtained 2,724,621,233 smORFs. A large majority (84.7%) were singleton sequences. To reduce the incidence of false positives[36], we focused first on the 417 million non-singleton sequences. We hierarchically clustered these non-singleton smORFs at 90% amino-acid identity and 90% coverage, which resulted in 287,926,875 clusters, which we will henceforth refer to as families. Then, we constructed the smORFs catalog, which contains both non-singletons as well as any singleton that matches a family representative at 90% amino-acid identity and 90% coverage (rescued singletons, see Methods). The final smORF catalog contains 964,970,496 smORFs.

The samples in our dataset had been previously manually curated into 75 habitats[21], which we further grouped into 8 broad categories: mammal gut, anthropogenic, other-human, other-animal, aquatic, human gut, soil/plant, and other (Methods, Supplementary Data 2). Despite the large number of samples we have collected, rarefaction analysis indicates that smORF diversity is far from covered (Fig. 1c; Supplementary Fig. 2a).

Approximately half of GMSC families consist of only one sequence, but the size distribution of families is long-tailed, so the largest 12.2% of families already cover half of the 100AA smORFs (Fig. 1d).

## 43 million smORFs are high-quality

Predicting smORFs can result in a high rate of false positives. Thus, in addition to discarding non-homolog singleton predictions, we performed several in silico quality tests including estimating coding potential of families using RNAcode[37] and additionally matching genomic predictions to publicly available metatranscriptomic and metaproteomics data (see Methods). In total, 43,642,695 (4.5%) of the smORFs pass all in silico quality tests and have at least one match in transcriptional or translational data. We henceforth refer to these as high-quality predictions (Supplementary Figs. 3 and 4).

To assess the comprehensiveness of our catalog, we matched small proteins encoded by GMSC smORFs to the RefSeq database[38] and previously published human microbiome small protein family datasets[33]. Only 5.3% of smORFs in our catalog are homologous to these previously reported small proteins (Fig. 1e). On the other hand, our catalog contains more than 80% of these reference datasets. For smORFs of high-quality predictions, a higher proportion (8.7%) show homology with these reference datasets, but they only cover *circa* 20% of the reference datasets (Supplementary Fig. 5a). Hence the high-quality predictions produce a large number of novel small proteins with high confidence that are not present in other reference datasets, but as the available transcriptome and metaproteome datasets are limited, discarding non-high-quality predictions would result in a large loss of coverage.

To explore the functions undertaken by the small proteins encoded by the smORFs in our catalog, we searched the small protein families against the Conserved Domain Database (CDD)[39] using RPS-BLAST[40,41]. Only 6.1% of small protein families containing 86,694,259 smORFs (8.98%) were assigned CDD domains, compared to 35.2% of canonical-length proteins (greater than 100 amino acids)[20]. As expected, smORFs in high-quality predictions are twice as likely to be assigned a CDD domain (18.8%, $P$ value < $10^{-308}$, hypergeometric test).

## Even conserved small proteins lack functional annotations

Using MMSeqs2 taxonomy[42] we predicted the taxonomic origin of contigs and transferred that prediction to the smORFs (Methods). This process returned a prediction for 81.6% of the 100AA smORFs, with more than half (56.9%) being assigned to a genus or species (Fig. 2a). Note that we used the GTDB database[43], which does not include phage or microeukaryotes.

We next investigated the taxonomic breadth and conservation of smORFs[28,44]. Of the 96,721,815 small protein families with at least three members, more than half of them (52,550,829) are genus-specific (Fig. 2b). Among these genus-specific families, most are species-specific, accounting for 39.7% of the families included in the analysis.

Although in some cases, smORFs may be present in plasmids and other mobile elements, we reasoned that multi-genus families would be especially likely to be present in multiple habitats and involved in critical cellular functions[33]. As expected, multi-genus families are more common in multiple habitats than the entire set of families with at least three members even when differences in family size distributions are taken into account, but the difference is not large (61.8% vs. 57.5%; $P$ value < $10^{-308}$, due to the large number of datapoints, hypergeometric test). Furthermore, we traced the conserved Pfam domains of small protein families[45] (Supplementary Data 4). Multi-genus families are annotated with Pfam domains at a higher rate than the background of all families with at least three members (9.91% vs 8.15%; $P$ value < $10^{-308}$, hypergeometric test). Nonetheless, it is noteworthy that the vast majority have no detected Pfam domain and that a further 9.5% of those annotated, were annotated with Pfam domains of unknown functions (Fig. 2c).

We then focused on conserved families present in multiple phyla. We found a total of 2437 multi-phylum families present across all 8 broad habitat categories (Supplementary Data 5). Of these, only 752 families were annotated with Pfam CDs, of which 268 (35.6%) were associated with ribosomal proteins and 99 (13.2%) belonged to the Helix-turn-helix clan (Fig. 2d).

## Archaea harbor more smORFs proportionally than bacteria

To investigate the presence of smORFs in different microorganisms without sampling bias, we calculated the number of redundant smORFs per megabase pairs (Mbp) of assembled contigs, also named the density of smORFs[32,46].

Most of the genera with the highest density come from *Pseudomonadota*, *Bacillota A*, *Actinomycetota*, *Bacillota*, and *Bacteroidota* (Fig. 3a). However, when considering the density of phyla as a whole, interestingly, we found the density of archaeal phyla is higher than bacterial ones ($P_{Mann}$ = 2.2510$^{-3}$; Fig. 3b). Of the ten phyla with the highest smORF density, half are archaeal, despite the fact that only 18 archaeal phyla contained enough data to be analyzed compared to 131 bacterial ones (Fig. 3c, Supplementary Data 6). The phyla that produce the most smORFs per Mbp are *Desulfobacterota D* (362.87 smORFs per Mbp), *Undinarchaeota* (331.35 smORFs per Mbp), *Nanoarchaeota* (281.34 smORFs per Mbp), *Methylomirabilota* (241.37 smORFs per Mbp), and *Huberarchaeota* (241.05 smORFs per Mbp).

## Differences in functions for archaeal and bacterial small proteins

Given the higher densities of smORFs in Archaea, we investigated the functions and properties in archaeal and bacterial small proteins encoded by smORFs[47]. We compared the archaeal and bacterial small protein families with COG[48] annotation. Only 1.72% of the families are annotated with COGs, of which 4,747,223 families are from bacteria and 202,825 families are from archaea. The COG classes that belong to Information storage and processing account for the largest proportion of small proteins in both bacteria and archaea (Fig. 4a), which is consistent with that found by Wang et al.[44]. However, *circa* 17% of small proteins in bacteria and archaea are still annotated as COG classes which are poorly characterized.

Small proteins with transmembrane or secreted characteristics may be involved in cell communication[7]. We explored the transmembrane and secreted small proteins in archaea and bacteria (Methods). 15.3% of the families are predicted to be transmembrane (using TMHMM-2.0[49]) or secreted (using SignalP-5.0[50]), with archaeal families

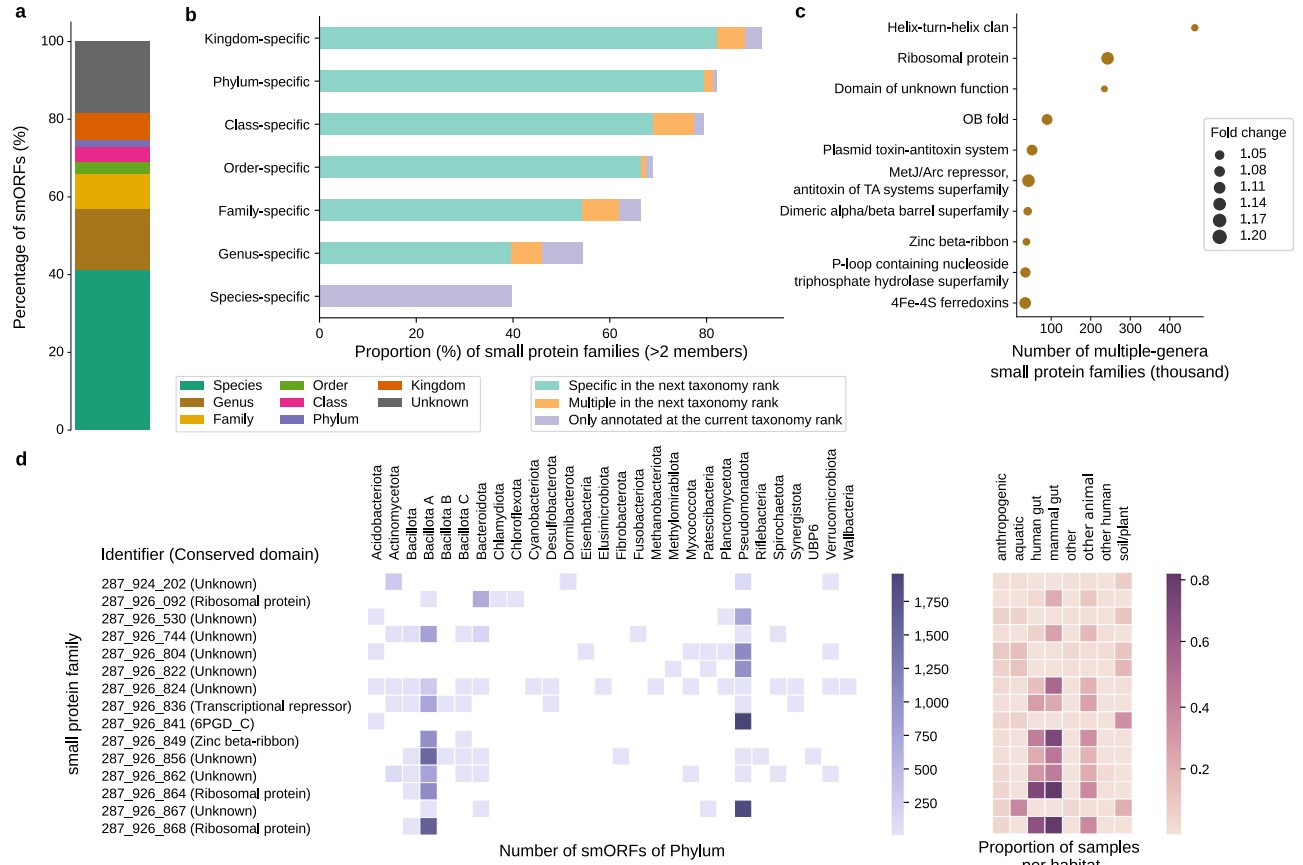

**Fig. 2 | Taxonomic and functional annotation of small proteins. a** Predicting taxonomy for the contigs and genomes from which smORFs originate (Methods) resulted in a taxonomic assignment for 81.6% of smORFs (56.9% of smORFs at genus or species level). **b** When only families with >2 members were considered (96,721,815 families), there are three cases at each taxonomic rank. For example, considering the rank of class, a small protein family is annotated to a particular taxonomic class if all its members are annotated as belonging to that class (unannotated smORFs being ignored). We further distinguish three cases, namely whether its members are (i, marked specific in the next taxonomic rank) all be annotated to the same order (as order is the next taxonomic rank), (ii, marked multiple in the next taxonomic rank) annotated to different orders within that class, being predicted at a higher rate than bacterial ones to be transmembrane or secreted ($P_{Mann}$ ≤ 0.0103, Fig. 4b)[51].

or (iii, marked only annotated at the current taxonomic rank) not annotated to any order. Other ranks are treated analogously (until we reach the level of species). **c** The enrichment of Pfam domains in small protein families present in multiple genera compared to the entire families with over two members (P value < 0.05, Hypergeometric Test, corrected by Bonferroni). Pfam domains were grouped by Pfam domain clans. Fold change is the ratio of the Pfam proportion of small protein families which present in multiple genera to the Pfam proportion of the entire families with over two members. **d** The Pfam annotation of small protein families that exist in multiple phyla, spanning >100 species and distributed across all the eight broad habitat categories (mammal gut, anthropogenic, other-human, other-animal, aquatic, human gut, soil/plant, and other).

Furthermore, compared with bacterial transmembrane or secreted small proteins, we found that archaeal transmembrane or secreted small proteins are enriched in COG classes related to the transport and metabolism of coenzymes, carbohydrates, and inorganic ions, besides the intracellular trafficking, secretion, and vesicular transport. In contrast, they are depleted in COG classes related to cellular processes and signaling (P value < 0.05, Fisher's exact test, multiple tests corrected by Bonferroni, Fig. 4c).

Some COGs were primarily (or even exclusively) present in archaea (as defined by a P value < 0.05, Fisher's exact test, multiple tests corrected by Bonferroni, Fig. 4d). For example, the COG with the highest proportion in archaea, COG4023 is a preprotein translocase subunit Sec61beta, which is a component of the Sec61/SecYEG protein secretion system. It is found in eukaryotes and archaea and is possibly homologous to the bacterial SecG[52].

### Identification of smORFs by GMSC-mapper
As mentioned above, smORF predictions are prone to false positives and one strategy for increasing confidence is to find sequences present in multiple genomes (or metagenomes). In this context, our catalog can be a resource whereby users with a single sample (or a small number of samples) use it as a reference to obtain high-quality predicted smORFs. For this usage, we provide a tool called GMSC-mapper (Fig. 5a).

GMSC-mapper performs de novo prediction and annotation of small proteins encoded by smORFs in user-provided genomes or assembled metagenomes (Methods). For this, it first uses Pyrodigal[35,53] to predict small proteins from assembled contigs and then it uses DIAMOND[54] or MMseqs2[55] to align these predictions against the GMSC. To minimize computational resource usage, the GMSC-mapper only searches family representatives, but it returns the set of matching smORFs and the annotation of the matches (e.g., habitat and taxonomy) as well as links to GMSC identifiers.

We compared DIAMOND to MMseqs2 for this task and observed that DIAMOND is faster than MMseqs2 when the number of query sequences is below 10,000, while MMseqs2 is slightly faster than DIAMOND when the number of queries is above 10,000 (Fig. 5b). In addition, we compared the number of recovered sequences (Fig. 5c) with either of these tools or BLAST[56], by randomly modifying sequences in the catalog and aligning these modified versions back to the catalog of family representatives. All three tools can find a high-identity match if it is present in the database. With increasing sequence

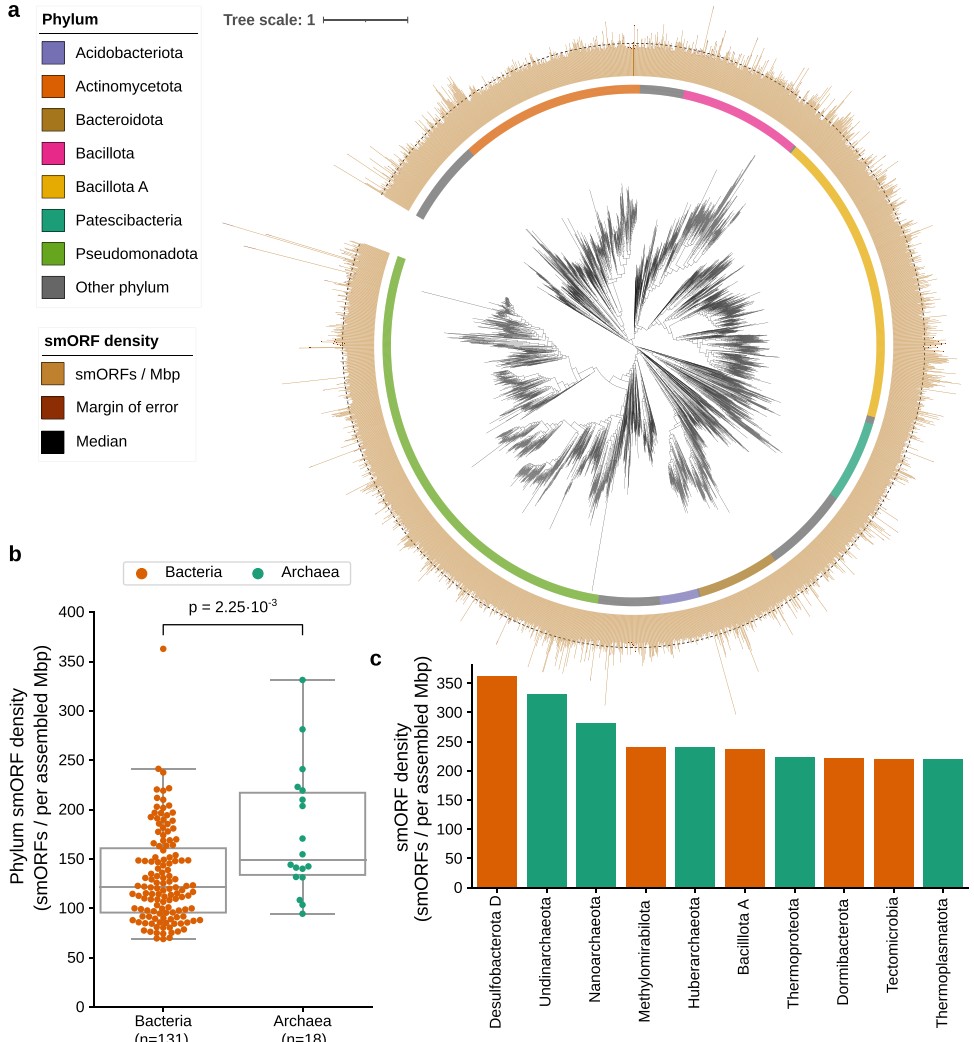

**Fig. 3 | Archaea harbor more smORFs than bacteria. a** Shown is the smORFs density distribution for the top 3000 bacterial genera with the highest density (brown bars, confidence interval of 95% shown as dark brown bars). Most of the densest genera are from *Pseudomonadota*, *Bacillota A*, and *Actinomycetota*. For reference, the black dashed line represents the median smORFs density for the presented genera. **b** Calculating the smORFs density of each phylum, the density of archaea is significantly higher than that of bacteria. Box plots indicate median (middle line), 25th, 75th percentile (box) and 5th and 95th percentile (whiskers) as well as outliers (single points) that lie within 1.5 IQRs of the lower and upper quartile. *P* values shown are from the Mann–Whitney test (two-sided). **c** The top 10 phyla with the highest smORF density are shown.

size, these tools can match more distant homologous sequences. In this case, DIAMOND achieves almost the same sensitivity as BLAST and is superior to MMseqs2[57].

However, independently of the method used, when the sequences are too short (20 amino acids), the rate of recovery decreases drastically. Fundamentally, for short sequences in a large database, even an identical match has a high likelihood of arising by chance[58,59]. This will manifest itself in a high *E* value[60] for true positives, making it impossible to distinguish false and true positive matches based on sequence comparisons alone. Therefore, while the use of a higher *E* value threshold will recover a larger fraction of true matches (>80% recovered with DIAMOND using $10^{-3}$ compared to *circa* 40% using $10^{-5}$, see Fig. 5c, d), the false discovery rate (FDR) will also increase[58].

## Discussion

Here, we constructed the global microbial smORFs catalog (GMSCv1, in its first version), which contains ~1 billion smORF sequences, of which 43 million are high-quality predictions, representing a large increase in the number of smORF sequences previously reported and serving as a resource for the microbiome research community. For each smORF and small protein family, we provide comprehensive annotations, including taxonomy, habitats, and CDs. Previously, most of the widely studied microbial small proteins were accidentally discovered in isolated and cultured bacterial species[5]. The large-scale discovery of small proteins has made great progress in recent years. Sberro et al.[33] conducted the characterization of conserved small proteins in the human microbiome, revealing their potential various functions. In our work, we have expanded the discovery of small proteins to 75 distinct habitats worldwide. In our catalog, only a small fraction are homologous to reference small protein datasets, with the vast majority of the novel small proteins being found in non-human-associated habitats (Supplementary Fig. 5b). On the other hand, it encompasses most of the known small proteins in either the RefSeq database or in families discovered recently (NMPfamsDB[61] and FesNov families[28]). When comparing with small protein databases that focus on eukaryotic organisms, such as smProt2[62], OpenProt2.0[63], and sORF.org[64], the overlap is minimal (Supplementary Fig. 5c).

A major difficulty in finding biologically functional smORFs is that false positive predictions are common. One of the underlying principles in our efforts is that finding identical or highly similar sequences in

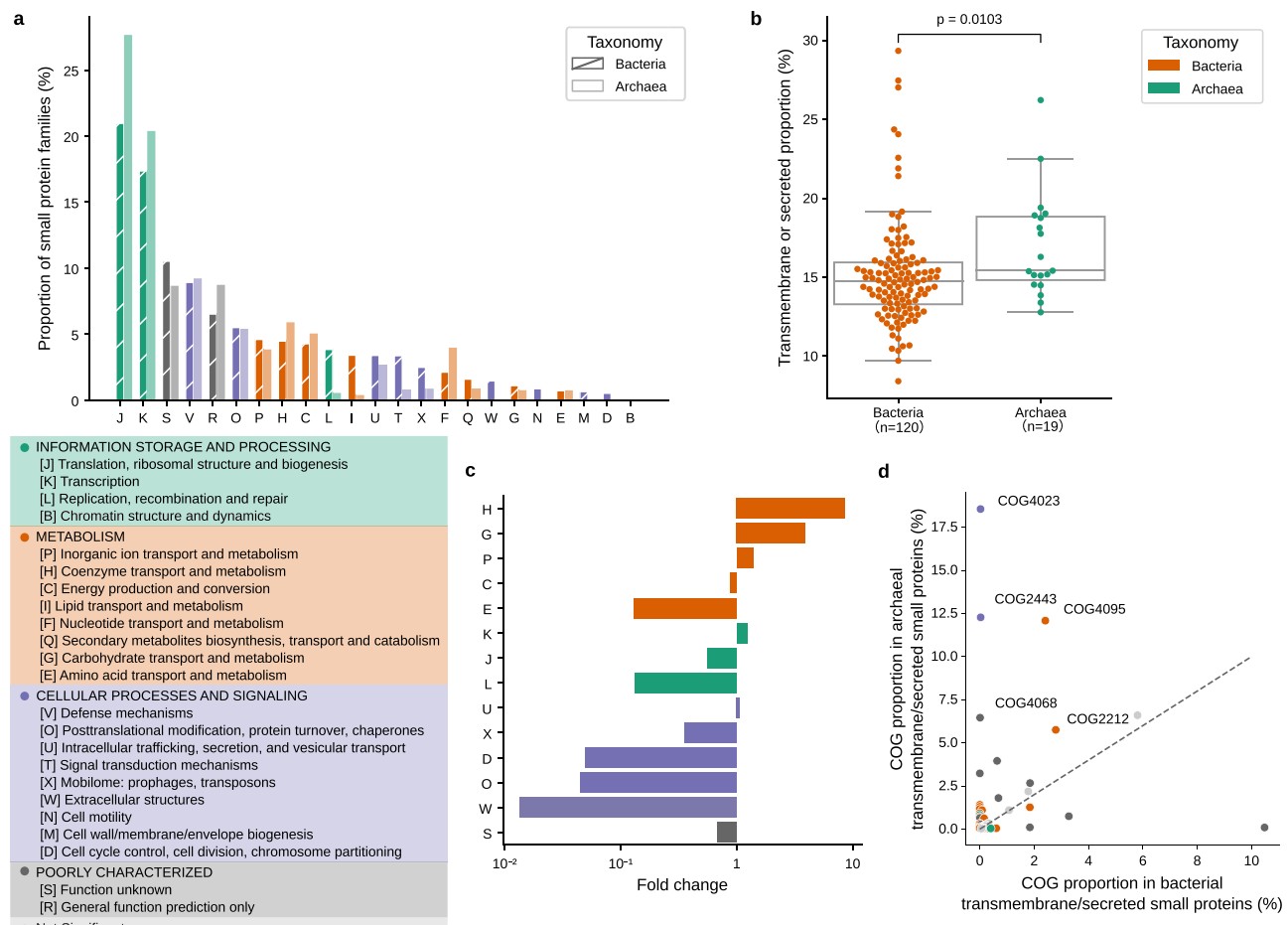

**Fig. 4 | Differences in functional prediction for archaeal and bacterial small proteins. a** The COG distribution of archaeal and bacterial small proteins is shown. **b** Archaea contain a higher fraction of transmembrane or secreted small proteins than bacteria (calculated per phylum). Box plots indicate median (middle line), 25th, 75th percentile (box) and 5th and 95th percentile (whiskers) as well as outliers (single points) that lie within 1.5 IQRs of the lower and upper quartile. *P* values shown are from the Mann–Whitney Test (two-sided). **c** Shown is the difference in the proportion of COG class in archaeal transmembrane or secreted small proteins versus bacterial transmembrane or secreted small proteins. The fold change is the ratio of proportions. The *P* values were calculated using Fisher's exact test (two-sided) and adjusted by Bonferroni correction. **d** Dots represent 43 COGs, which are enriched in archaeal transmembrane or secreted small proteins compared to the archaeal small proteins that are not transmembrane or secreted, as well as bacterial transmembrane or secreted small proteins. The proportion comparison of these 43 COGs between archaeal transmembrane or secreted small proteins and bacterial transmembrane or secreted small proteins is shown.

multiple samples increases the likelihood of a true prediction. Therefore, we discarded singleton predictions in our data. This principle also underlies the GMSC-mapper tool, which enables users to find matches from their datasets in GMSCv1.

As previously done[33], we have only conducted RNAcode[37] on small protein families with at least eight members to identify smORFs families with transcription signatures. This approach may, however, fail to identify some rapidly evolving functional smORFs. In addition, given the limited size and number of existing datasets of metatranscriptomes, (meta)Ribo-Seq and metaproteomes, the high-quality predictions are expected to underestimate the true diversity.

Computing approaches and concepts developed over decades for longer proteins do not necessarily work well for small sequences. For example, for the alignment of very short sequences, the minimum achievable *E* value will be lower bounded[60]. Even an identical match will obtain a relatively high *E* value as short identical matches can occur by chance. Furthermore, traditional databases lack small proteins, so functional assignment by orthology or with HMMs only returns a prediction for a minute fraction of all small proteins. We lack functional predictions for most small proteins in our dataset, even for those small protein families that are ubiquitous. Similarly, tools for predicting whether proteins are transmembrane or secreted are not

optimized for small proteins and our results should be interpreted in this context. In particular, when we compared results between bacteria and archaea, we implicitly assumed that the methods have similar error rates in these two domains, but this may not be the case. In related work, we used machine learning[36] to identify candidate antimicrobial peptides (AMPs) from the GMSC[46]. However, functional prediction for small proteins remains an open challenge, open to new approaches.

Overall, our resource shows the immense and underexplored diversity of small proteins across different habitats and taxonomy, and highlights the gaps in our scientific knowledge, while constituting a resource for the research community.

## Methods

### Collection of global metagenomic datasets and prediction of smORFs

In total, 63,410 publicly available global assembled metagenomes from the SPIRE database[21] collection were used. Briefly, the assembled metagenomes have been generated through the following methods: publicly available data (as of 1 January 2020) were downloaded from the European Nucleotide Archive (ENA) and short reads that were at least 60 bps after trimming positions with quality <25[65] were

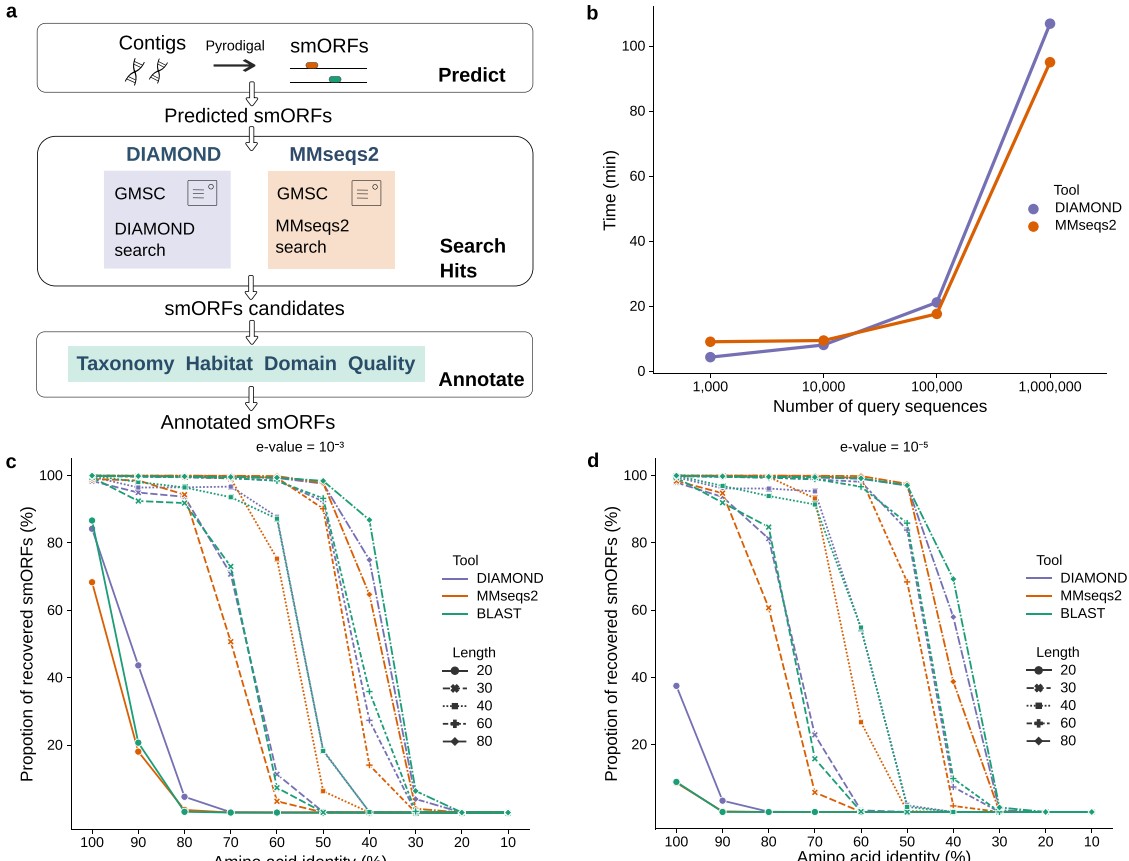

**Fig. 5 | Workflow and benchmark of GMSC-mapper. a** GMSC-mapper uses Pyrodigal to predict small proteins with <100 amino acids from contigs. Users can alternatively provide smORF or protein sequences directly, skipping the initial step of gene prediction. DIAMOND or MMseqs2 are used for finding homologs within GMSC. In the end, GMSC-mapper combines all alignment hits and provides detailed annotations of small proteins. **b** Time cost tests were performed among different numbers of input sequences from 1000 to 1,000,000 using DIAMOND and MMseqs2 (Methods). We compared the number of recovered sequences with different lengths (20, 30, 40, 60, and 80 amino acids) at different amino acid identities from 10% to 100% using DIAMOND, MMseqs2, and BLAST (Methods). The recovered number is influenced by the $E$ value cutoff used ($10^{-3}$ in **c** and $10^{-5}$ in **d**).

assembled into contigs using MEGAHIT 1.2.9[66]. Additionally, we downloaded 87,920 high-quality isolate microbial genomes from the ProGenomes2 database[34].

We then used the modified version of Prodigal[35] in Macrel 0.5[36] to predict ORFs ≥ 30 base pairs (bps) on the assembled contigs as well as those from Progenomes2 database. This version of Prodigal uses the same algorithm as the standard version of Prodigal, but with a lower limit on the size of genes. We used command line parameters to only predict closed genes, to not predict genes with N as a base, to perform a full motif scan, in metagenomics mode (-c -m -n -p meta). The ORFs encoding small proteins (here defined as those up to 100 amino acids) were considered smORFs.

We recorded the habitats of smORFs according to their source samples using the habitat microontology introduced in SPIRE database[21]. We further grouped the habitats into 8 broad categories: mammal gut, anthropogenic, other-human, other-animal, aquatic, human gut, soil/plant, and other. We used GeoPandas[67] to present geographic coordinates of samples.

### Non-redundant smORFs catalog construction and method validation

All the smORFs were first deduplicated at 100% amino-acid identity and 100% coverage. Then we hierarchically clustered the non-singletons at 90% amino-acid identity and 90% coverage using Linclust[55,68] with the following parameters: -c 0.9, –min-seq-id 0.9. Linclust is a single-linkage approach, whereby sequences are clustered together if they share a common representative with candidate representatives being chosen heuristically.

Of these clusters, 47.5% contain a single sequence (singleton clusters). To rule out the possibility that this was due to the fact that Linclust[55,68] is a heuristic method that is not specifically designed for short sequences, we estimated the rate of false negatives (i.e., sequences that were marked as singleton even though they should have been clustered with another one). We aligned a randomly selected 1000 singleton clusters against the representative sequences of non-singleton clusters (i.e., those containing ≥2 sequences) using SWIPE[69] with the following parameters: -a 18 -m '8 std qcovs' -p 1. The alignment threshold was $E$ value $< 10^{-5}$, identity ≥90%, and coverage ≥90% (Supplementary Fig. 1a).

In addition, to estimate the rate of false positive clusterings (sequences that were assigned to a cluster even though they do not share the required identity with the cluster representative), 1000 sequences were randomly selected and aligned against the representative sequences of their clusters using SWIPE[69] with the following parameters: -a 18 -m '8 std qcovs' -p 1. The alignment threshold was $E$ value $< 10^{-5}$, identity ≥90%, and coverage ≥90% (Supplementary Fig. 1b).

When clustering, we initially discarded the singletons because singletons are enriched in artifactual smORFs[36]. However, we considered that singletons that are homologous to larger clusters should not be discarded as the homology itself provides further evidence of biological relevance. Therefore, we aligned singletons to the

representative sequences of clusters with 90% sequence identity and 90% coverage using DIAMOND[54] using parameters: -e $10^{-5}$ –id 90 -b 12 -c 1 –query-cover 90 –subject-cover 90. We combined the homolog singletons and the non-singleton sequences identified earlier and termed them the smORFs catalog containing 964,970,496 smORFs.

## Sample-based smORFs rarefaction curves
Samples were randomly permuted 24 times to calculate the total number of non-redundant smORFs captured as the number of samples increased. We took the average across the permutations as the final estimate.

## Quality control of the catalog
We conducted several in silico quality tests and matched genomic predictions to other publicly available experimental data.

A smORF predicted at the start of a contig that is not preceded by an in-frame STOP codon risks being a false positive originating from an interrupted fragment. Therefore, we checked for the presence of an upstream in-frame STOP. For smORFs without an upstream in-frame STOP, however, we could not determine whether there were other genes present upstream of them (Supplementary Fig. 3a).

To avoid spurious smORFs, we used HMMSearch[70] with the --cut_ga option to search smORFs against the AntiFam 7.0 database[71], which contains a series of confirmed spurious protein families.

We used RNAcode[37], a tool to predict the coding potential of sequences based on evolutionary signatures, to identify the coding potential of 25,744,932 smORF families containing ≥8 sequences. The smORF families with P value < 0.05 were considered to have coding potential, as in a previous study[33] (Supplementary Fig. 4a).

Furthermore, we searched for evidence that these smORFs are transcribed and/or translated. For this step, we downloaded 221 publicly available metatranscriptomic datasets from the NCBI database paired with the metagenomic samples we used in our catalog (Supplementary Data 3). These samples are from the human gut, peat, plant, and symbionts. To keep the procedure computationally feasible, we mapped reads against the representative sequences of smORF families by BWA[72]. Then we used NGLess[65] with 'unique_only' for the 'multiple' argument of the count built-in function to only count uniquely mapped inserts. A smORF family was considered to have transcriptional evidence if its representative has reads mapped to it in at least 2 samples (Supplementary Fig. 4b). Furthermore, we mapped reads against the smORFs in paired metagenomic and metatranscriptome samples, separately. On average, 58.6% of the smORFs in each paired sample are mapped.

We downloaded 142 publicly available Ribo-Seq datasets from the NCBI database (Supplementary Data 3). We also mapped reads against representative sequences of smORF families by BWA[72]. Then we used NGLess[65] with 'unique_only' for the 'multiple' argument of the count built-in function to only count uniquely mapped inserts. A smORF family was considered to have translation evidence only if its representative has reads mapped to it in at least 2 samples (Supplementary Fig. 4c).

Moreover, we downloaded peptide datasets from 108 metaproteomic projects from the PRIDE database[73] (Supplementary Data 3). We matched GMSC smORFs to the identified peptides of each project. If the total k-mer coverage of peptides on a smORF is greater than 50%, then the smORF is considered translated and detected, as in a previous study[74] (Supplementary Fig. 4d).

Sequences that passed all in silico tests above as well as matching transcriptional or translational data were regarded as high-quality predictions.

## Comparison with reference small protein datasets
We downloaded bacterial and archaeal protein sequences from RefSeq in March 2023[38], consensus sequences of NMPFamsDB[61] and

sequences for each FESNov gene family[28]. The sequences with fewer than 100 amino acids are considered small proteins, and redundancy was subsequently removed with 100% amino-acid identity and 100% coverage. A total of 16,333,323 bacterial small proteins, 368,769 archaeal small proteins from RefSeq, 56,786 small proteins from NMPFamsDB, and 630,375 small proteins from FESNov families were included in the comparison. We also included the 444,053 small protein clusters and 4539 conserved small protein families from Sberro's human microbiome study[33]. We compared our smORFs catalog to these datasets using DIAMOND with the '–more-sensitive' mode, retaining significant hits (E value < $10^{-5}$). In addition, we compared our smORFs catalog with small protein sequences provided in current small protein database mainly about eukaryotic organisms. We downloaded small proteins from human, mouse, yeast, rat, *E. coli*, *C. elegans*, fruitfly, zebrafish, and small proteins from LiteratureMining, KnownDatabase, and MSfragments from SmProt2 database[62]; small proteins from human, mouse, rat, zebrafish, fruitfly, *C. elegans* of sORF.org database[64]; and all predicted refprots, altprots, and isoforms sequences with all annotations from human, chimp, rat, mouse, zebrafish, fruitfly, *C. elegans*, and yeast from OpenProt2.0 database[63]. After filtering small proteins by length and removing redundancy as above, 788,586 small proteins from SmProt2 database, 4,377,422 small proteins from sORF.org database, and 1,781,907 small proteins from OpenProt2.0 database were included in the comparison. As above, we compared our smORFs catalog to these datasets using DIAMOND with the '–more-sensitive' mode, retaining significant hits (E value < $10^{-5}$).

## Conserved domain annotation
We downloaded the CDD[39] from ftp://ftp.ncbi.nih.gov/pub/mmdb/cdd/little_endian/Cdd_LE.tar.gz in September 2022, which contains models from CD curated at NCBI, Pfam[45], SMART[75], COGs[48], PRK[76], and TIGRFAMs[77]. All the representative sequences of small protein families were searched against the CDD by RPS-BLAST[40,41]. In order to establish a comparison baseline, we additionally randomly selected 10,000 prokaryotic proteins from the global microbial gene catalog v1.0[20] and searched them against the CDD by RPS-BLAST[40,41]. Hits with an E-value maximum of 0.01 and at least 80% of coverage of PSSM's length were retained and considered significant. Pfam accessions were grouped by Pfam clan[78] or the first phrase before the comma in their short description.

## Taxonomic annotation and taxonomic breadth analysis
The taxonomy of assembled contigs encoding the small proteins was annotated using MMseqs2 taxonomy[42] against the GTDB database[43] release r95. However, in figures and text, we used updated taxon names (e.g., *Bacillota* instead of *Firmicutes*). We characterized the taxonomy of predicted smORFs based on the taxonomy of contigs and microbial genomes[34] from which the smORFs were predicted. We subsequently assigned taxonomy for GMSC smORFs and families using the lowest common ancestor, ignoring the un-assigned ranks to make them more specific.

The small protein families with at least three members were subsequently used to perform taxonomic breadth analysis. Each family was classified according to (i) whether it is single or multi-habitat; (ii) whether it is single or multi-genus; and (iii) whether it is annotated with a Pfam domain[45]. Multi-genus families are more common in multiple habitats than the entire families (61.8% vs. 52.0%; P value < $10^{-308}$, hypergeometric test). Multi-genus families are annotated with Pfam domains at a higher rate (9.91% vs 7.52%; P value < $10^{-308}$, hypergeometric test). As these results could have been confounded by differences in family size distributions, we randomly downsampled the data to keep the same number of families at each size between multi-genus families and the whole families. In that case (as presented in the main text), the difference was 61.8% vs. 57.5% (P value < $10^{-308}$, hypergeometric test) for the proportion of families in multiple habitats and

9.91% vs. 8.15% ($P$ value $< 10^{-308}$, hypergeometric test) for the proportion of Pfam annotated families.

## Density calculation

The density of smORFs was defined as $\rho = n_{smORFs} / L$, where $n_{smORFs}$ is the number of redundant smORFs and L is the assembled megabase pairs (Mbps)[32,46]. The density was calculated by summing all assembled base pairs for contigs assigned to each taxonomic rank. We assume a scenario where the starting positions of smORFs in an assembled large contig are independent and uniformly random. Therefore, the standard sample proportion error was calculated as $STD_{err} = \sqrt{\frac{\rho*(1-\rho)}{L}}$ and was used to calculate the margin of error at a 95% confidence interval (Z = 1.96). We did not further consider the calculated values with a margin of error >10%.

## Cellular localization prediction

To detect potential transmembrane proteins, we ran TMHMM-2.0[49] on the representative sequences of small protein families. Then, to identify potentially secreted small proteins, we used SignalP-5.0[50] on the representative sequences of small protein families. For families classified as archaea, we used '-org arch', while for the others we combined the outputs of '-org gram + ' and '-org gram-' modes.

## Construction and evaluation of GMSC-mapper

GMSC-mapper supports assembled contigs, smORF sequences, or protein sequences as inputs. It uses Pyrodigal[35,53], which is a faster implementation of the Prodigal algorithm, to predict ORFs potentially coding for small proteins (those with fewer than 100 amino acids) from contigs. Gene prediction is skipped when inputs are smORF or protein sequences. Then DIAMOND[54] or MMseqs2[55] are used for homologous alignment against GMSC. Finally, it combines all the alignment hits information and provides detailed annotation of small proteins.

To determine the optimal default sensitivity mode, we tested different sensitivity parameters for DIAMOND and MMseqs2. We aligned 10,000 randomly selected sequences back to the family representatives and counted the number of recovered sequences while monitoring the computational time. We use the "–sensitive" mode as the default sensitivity parameter for DIAMOND, which provides the best balance between sensitivity and speed. The use of more-sensitive modes resulted in little or almost no increase in the number of recovered sequences, but a substantial increase in time usage. For MMseqs2, we keep the original default sensitivity parameter (5.7) considering that the number of recovered sequences and the time both increase with the increase of sensitivity (Supplementary Fig. 6a-d).

We then tested the time costs among different numbers of input sequences using the "–sensitive" mode of DIAMOND and the default sensitivity parameter (5.7) of MMseqs2. GMSC-mapper can annotate 100,000 input sequences in approximately 20 minutes with 20 threads.

Furthermore, we compared the number of recovered sequences with different identities using different alignment tools. We randomly selected and mutated 10,000 sequences of different lengths (20, 30, 40, 60, and 80) from the family representatives, with different identities from 10% to 90%. We aligned them back to the family representatives using DIAMOND, MMseqs2, and BLAST[56], respectively. When the query sequence and the target sequence are the same, we consider them as the recovered sequences.

Timing measurements were performed using a server equipped with an AMD EPYC 7763 64-Core processor and 2TB of RAM memory.

## Statistics and reproducibility

Statistical analyses were carried out in Python 3.8.5, using Pandas[79] 1.1.3, NumPy[80] 1.24.4, and SciPy[81] 1.10.1. No statistical method was used to predetermine sample size. No data were excluded from the analyses. The experiments were not randomized. The investigators were not blinded to allocation during experiments and outcome assessment.

## GMSC web resource

GMSC webserver is hosted at the address https://gmsc.big-data-biology.org, where an implementation of GMSC-mapper can be accessed. The website implementation is based on Elm-Lang. The API implementation is based on Python.

## Reporting summary

Further information on research design is available in the Nature Portfolio Reporting Summary linked to this article.

## Data availability

Global metagenomic data are publicly available at the ENA. The accession numbers for samples and studies are listed in Supplementary Data 1. Microbial genomes are publicly available in the Progenomes2 database. The global microbial smORFs catalog (GMSC) resource has been deposited in Zenodo under https://doi.org/10.5281/zenodo.7944370. The resource is freely available at https://gmsc.big-data-biology.org. Users can query small protein sequences by using GMSC-mapper through the web interface or select their interesting small proteins by habitats and taxonomy.

## Code availability

The codes used to generate and analyze the global microbial smORFs catalog (GMSC) are available at https://github.com/BigDataBiology/Duan2024_GMSCv1_Construction_And_Analysis, archived at Zenodo under https://doi.org/10.5281/zenodo.13119583. GMSC-mapper is open source and at https://github.com/BigDataBiology/GMSC-mapper.

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

## Acknowledgements

This work was supported by the National Natural Science Foundation of China (T2225015, 61932008) (L.P.C., Z.X.M.), Shanghai Science and Technology Commission Program (23JS1410100) (L.P.C., Z.X.M.), National Key R&D Program of China (2023YFF1204800, 2020YFA0712403) (L.P.C., Z.X.M.), Key Science and Technology Project of Hainan Province (ZDYF2024SHFZ058) (Z.X.M.), Major Project of Guangzhou National Laboratory (GZNL2024A01003) (Z.X.M.), Lingang Laboratory & National Key Laboratory of Human Factors Engineering Joint Grant (LG-TKN-202203-01) (Z.X.M.), and the Australian Research Council (grant FT230100724). We thank Ben Woodcroft (Queensland University of Technology) for helpful comments on a previous version of the manuscript.

## Author contributions

L.P.C. conceptualized and designed the study. Y.D., C.D.S.J., T.S.B.S., A.F., L.P.C., M.K. curated the data. Y.D., C.D.S.J., B.L.S.D., and C.Z. analyzed and visualized the data. L.P.C., X.M.Z., and P.B. supervised the project. Y.D. and L.P.C. wrote the original draft. All authors reviewed and contributed to the manuscript.

## Competing interests

The authors declare no competing interests.
