## [Peer Review File · Nature Communications]

A catalogue of small proteins from the global microbiomeREVIEWER COMMENTS

Reviewer #1 (Remarks to the Author):

Duan et al. report a curated collection of small proteins from across bacterial genome diversity. The study is well conducted and the resource will be useful for prokaryotic genome annotation. I only have a few minor comments that I think would be useful to implement, at the authors' discretion.

I think the databases NMPfamsDB and SmProt should be mentioned, and the differences to GMSC discussed. I recognise that NMPfamsDB has only very recently been released (and that there is overlap in authors) but I think it could now be useful to include some mention of it.

It may be useful to discuss or mention the fact that many of the ORFs may be from mobile genetic elements such as plasmids or phages.

I would like to see some mention of why the key thresholds (e.g. $p < 0.05$ for RNACode) were chosen. I appreciate however that space is limited.

Similarly, some more detail of how Prodigal was used could be useful.

Other comments

Supplementary Fig 3b: label should say "not performed"

~ line 282: I did not find the description of the terminal checking very clear here. Perhaps a diagram could be provided?

Line 284 - the version of Antifam used could be stated. <https://interpro-documentation.readthedocs.io/en/latest/antifam.html>

Line 398 - “were carried out”

Reviewer #1 (Remarks on code availability):

I have briefly surveyed the code provided.

It appears to be well written, and providing code for all figures as jupyter notebooks, with associated data appropriately organised, is excellent practice.

Reviewer #2 (Remarks to the Author):

The manuscript entitled: “A catalogue of small proteins from the global microbiome” provides a pipeline to generate a large-scale catalogue of smORFs in microbes. The library has the potential to serve as a resource for the microbiome community.

General comments:

The manuscript provides a potentially useful resource of putative microbial smORFs. The resource is fairly well characterized and investigated. However, the manuscript is not easy to follow in terms of how it presents both its methods and results. Many method descriptions are in the Results section or figure captions, while many results are reported in the Methods section. This makes for a hard read of the manuscript. Furthermore, very few criteria used for building this catalogue are justified. The majority of criteria are arbitrary thresholds selected by the authors. Finally, little is done to convince the reader about the validity of the catalogue and its potential use. I provide more specific comments below.

Major comments:

A large number of results are provided in the Methods section. These included, but are not limited to, results provided at line 257, 262, 269, 281, 286, 289, 299, 308, and 313. On the other hand, methods are often described better or repeated in figure captions. A significant reorganization of the text is needed to ease reading.

Moreover, criteria and thresholds for metatranscriptomics, ribo-seq, and metaproteomics smORF confirmation are quite arbitrary. Instead of reporting a single number of confirmed smORFs at a selected threshold. Plots showing the number of smORFs passing at varying thresholds would provide a better grasp of the dataset, and would help providing a reasoning behind the choices of the different thresholds.

The smORFs family construction requires a more detailed explanation. What are the sequences that are clustered? Is it that any sequence that have at least one other sequence with which it has a 90% identity and 90% coverage is used as input for the clustering analysis? You could have three sequences named, A, B and C, with A and B having a 90% identify, A and C also having a 90% identity, but B and C not having this level of identity. Would these be all grouped together?

The procedure used to evaluate the significance of the clusters appears convoluted, under sampled and arbitrary. Why not using a simple bootstrapping approach to evaluate the robustness of the clusters? This is a lot more standard and typical for such analyses. Also how are the representative sequences of the clusters determined? This is not clearly described in the methods.

How does the proposed catalogue compare to that of OpenProt and the method they used for their database construction (Sébastien Leblanc, Feriel Yala, Nicolas Provencher, Jean-François Lucier, Maxime Levesque, Xavier Lapointe, Jean-Francois Jacques, Isabelle Fournier, Michel Salzet, Aïda Ouangraoua, Michelle S Scott, François-Michel Boisvert, Marie A Brunet, Xavier Roucou, OpenProt 2.0 builds a path to the functional characterization of alternative proteins, *Nucleic Acids Research*, Volume 52, Issue D1, 5 January 2024, Pages D522–D528, <https://doi.org/10.1093/nar/gkad1050>). OpenProt did not

cover bacteria so far and they do not have the same objectives, but some of them are overlapping. Would their approach be applicable here?

The data does not back-up the following conclusion stated by the authors: " Archaea have more transmembrane or secreted small proteins than bacteria". First, no evidence is provided that these specific smORFs are translated into proteins. Second, archaea have way less data points and I would assume that if one would remove a couple of the highest points that look more like outliers than anything else, the result would not be significant anymore. This result appears to be an artefact of the methods used to identify smORFs and transmembrane domains more than anything else. The conclusion of this entire section should be removed or rewritten.

To provide further insights into the validity of the catalogue, I would have expected that sequence conservation would have been directly investigated. One would assume that high-quality predictions are more likely to be functional than low-quality ones. Hence, they should be more likely to be evolutionarily conserved. Is it the case that nucleotides part of these high-quality predictions are more conserved than those that are of lower quality. A fold-enrichment could be provided to yield such an assessment.

In order to provide some insights into the potential applications and discovery potential of the catalogue, it would be interesting to see how these novel smORFs can help identify more peptides and proteins in metaproteomics studies. Most mass spectrometry-based metaproteomics studies will identify proteins using a technique called sequence database search. Providing a set of smORFs not typically included in such sequence database searches could help reveal new proteins never identified in metaproteomics datasets in the past.

No ReadMe are provided with the code, making its evaluation extremely difficult.

Minor comments:

Why were 10,000 randomly selected prokaryotic proteins queried using RPS-BLAST?

Versions used should be provided for Python, Pandas, NumPy, and SciPy.

Reviewer #2 (Remarks on code availability):

The code was not thoroughly reviewed due to a lack of instructions on how to execute it.

Response to reviewer comments

We thank the reviewers and editors for their time and comments. We provide a detailed point-by-point response below, but the main changes in the current revision are:

1. We rewrote the conclusions in the section *Archaea have more transmembrane or secreted small proteins than bacteria* to acknowledge that the results are only based on predictions.
2. We show the results of using different quality thresholds. This is shown both in the manuscript (Supplementary Fig. 4) and in the updated version of the website.
3. We added a README file to the supporting code which we also reorganized and renamed for clarity.

Point-by point response

Reviewer #1 (Remarks to the Author):

Duan et al. report a curated collection of small proteins from across bacterial genome diversity. The study is well conducted and the resource will be useful for prokaryotic genome annotation. I only have a few minor comments that I think would be useful to implement, at the authors' discretion.

I think the databases NMPfamsDB and SmProt should be mentioned, and the differences to GMSC discussed. I recognise that NMPfamsDB has only very recently been released (and that there is overlap in authors) but I think it could now be useful to include some mention of it.

Author response: We appreciate the reviewer's suggestions for comparing our results with recently published microbial protein databases. In addition to NMPfamsDB and SmProt2, we also compared our catalogue with the FESNov protein families, which contain previously uncharacterized genes from uncultivated taxa (del Río et al., 2023; <https://www.nature.com/articles/s41586-023-06955-z>), OpenProt2.0 (Leblanc et al., 2023; <https://doi.org/10.1093/nar/gkad1050>) and sORF.org (Olexiouk et al., 2017; <https://doi.org/10.1093/nar/gkx1130>). With respect to NMPfamsDB and FESNov (which are microbially-focused), our catalogue encompasses most of the small proteins they report. However, we found that our microbial small proteins have very limited overlap with SmProt2, OpenProt2.0, and sORF.org databases which mainly contain small proteins from eukaryotic organisms.

Changes made: We added a new Supplementary figure panel (**5c, see below**) showing the overlap with these databases. We also mention the comparison with small proteins from NMPfamsDB, FESNov protein families, SmProt2, OpenProt2.0, and sORF.org database in Line 233 of Discussion, as below "On the other hand, it encompasses most of the known small proteins in either the RefSeq database or in families discovered recently (NMPfamsDB and FesNov families). When comparing with small protein databases that focus on eukaryotic organisms, such as smProt2, OpenProt2.0, and sORF.org, the overlap is minimal (Supplementary Fig. 5c).

Supplementary Fig. 5 (reproduced here for convenience, compared to the previous version, panel c was added; note that this was previously Sup. Fig. 4). Comparison of reference small protein datasets (a) Shown is the fraction of smORFs from high-quality predictions that are homologous to reference small protein datasets. **(b)** The comparison of the proportions of smORFs from human or non-human habitats between homologs or non-homologs to small protein clusters and conserved families from the Sberro human microbiome dataset. **(c)** Shown is the fraction of GMSC smORFs that are homologous to NMPfamsDB, FesNov families, smProt2, OpenProt2.0, and sORF.org.

It may be useful to discuss or mention the fact that many of the ORFs may be from mobile genetic elements such as plasmids or phages.

Author response: This is an excellent point. As we now make explicit, when estimating taxonomy we mapped to the GTDB database which only includes prokaryotic genomes. We further mention the possibility that some ORFs may be part of mobile elements in the Results Section “Even conserved small proteins lack functional annotations”.

Changes made: We added the following sentence in Line 126 of Results: “Note that we used the GTDB database, which does not include phage or microeukaryotes.” We rewrote

the sentence in Line 131 of Results: “Although in some cases, smORFs may be present in plasmids and other mobile elements, we reasoned that multi-genus families would be especially likely to be present in multiple habitats and involved in critical cellular functions.”

I would like to see some mention of why the key thresholds (e.g. $p < 0.05$ for RNAcode) were chosen. I appreciate however that space is limited.

Author response and changes made: We followed the RNAcode threshold standard used by Sberro et al. (<https://doi.org/10.1016/j.cell.2019.07.016>). In the revised version, we measured the number of coding-potential smORFs under different p-value thresholds of RNAcode in Supplementary Fig. 4a. Now we mentioned that in Line 348 of Methods: “The smORF families with p-value < 0.05 were considered to have coding-potential, as in a previous study (<https://doi.org/10.1016/j.cell.2019.07.016>) (Supplementary Fig. 4a).”

Supplementary Fig. 4 (reproduced here for convenience). Effect of different thresholds on quality control (a) The number of smORFs with high coding potential as estimated by RNAcode, using different P-value thresholds. **(b)** The number of smORFs with transcriptional evidence, using different thresholds for the minimal number of samples required for detection. **(c)** The number of smORFs with translational evidence, using different thresholds for the minimal number of samples

required for detection. **(d)** The number of detected smORFs in metaproteomics data, using different thresholds for the required k-mer coverage of each smORF-encoded small protein (Methods).

Similarly, some more detail of how Prodigal was used could be useful.

Author response: As this was the default mode in the previously published Macrel tool that we cite, we had not elaborated in the previous version, but we now describe in detail the options used.

Changes made: In the corresponding Methods section, we added a brief description of the changes to Prodigal and, the command line parameters used (Line 278, novel text in bold): “We then used the modified version of Prodigal in Macrel 0.5 to predict open reading frames (ORFs) ≥ 30 base pairs (bps) on the assembled contigs as well as those from Progenomes2 database. **This version of Prodigal uses the same algorithm as the standard version of Prodigal, but with a lower limit on the size of genes. We used command line parameters to only predict closed genes, to not predict genes with N as a base, to perform a full motif scan, in metagenomics mode (-c -m -n -p meta).**”

Other comments

Supplementary Fig 3b: label should say “not performed”

Author response and changes made: We thank the reviewer and changed the label to “not performed” as follows below (this is now Supplementary Fig. 3c):

Supplementary Fig. 3 (reproduced here for convenience). Quality assessment workflow and overlap (a) To rule out the possibility that a smORF is part of a longer gene due to contig fragmentation, we searched for an in-frame STOP codon upstream of the smORF START. **(b)** The computational quality tests include (i) Terminal checking to reduce the risk that the smORF is derived

from a fragmented longer gene (as illustrated in **a**); (ii) AntiFam searches to avoid spurious protein families; and (iii) RNaCode estimated coding potential. The experimental data validation consists of mapping the metatranscriptomic and Ribo-Seq reads downloaded from the public database and exactly matching metaproteomic peptides downloaded from the Proteomics Identification Database (PRIDE). SmORFs were considered **_high-quality predictions_** if they passed all computational quality tests and were found in at least one experimental dataset. **(c)** Fraction of GMSC smORFs for each test. RNaCode was performed only on clusters with at least 8 members. Terminal checking was performed only on smORFs derived from metagenomes. **(d)** The upset plot shows the number of overlapping sequences passing each quality testing method.

line 282: I did not find the description of the terminal checking very clear here. Perhaps a diagram could be provided?

Author response and changes made: Thank you for the suggestion, we added a diagram as Supplementary Fig. 3a (as shown above). The process searches for an in-frame STOP codon upstream of the smORF to rule out the possibility that the smORF is part of a broken gene due to contig fragmentation.

Line 284 - the version of Antifam used could be stated.
<https://interpro-documentation.readthedocs.io/en/latest/antifam.html>

Author response and changes made: We used the latest version Release 7.0 of the AntiFam database, which has 263 entries, and added the following information to Line 343 (novel text in bold) of Methods: “To avoid spurious smORFs, we used HMMSearch with the **--cut_ga** option to search smORFs against the AntiFam **7.0** database, which contains a series of confirmed spurious protein families.”

Line 398 - “were carried out”

Author response and changes made: We have corrected the sentence in Line 480 of Methods: “Statistical analyses **were carried out** in Python 3.8.5, using Pandas 1.1.3, NumPy 1.24.4, and SciPy 1.10.1.”

Reviewer #1 (Remarks on code availability):

I have briefly surveyed the code provided. It appears to be well written, and providing code for all figures as jupyter notebooks, with associated data appropriately organised, is excellent practice.

Author response: We appreciate the reviewer's affirmation of the code of the manuscript. We have also updated the README document for data processing codes in catalogue construction to make it easier to understand and reproduce.

Reviewer #2 (Remarks to the Author):

The manuscript entitled: “A catalogue of small proteins from the global microbiome” provides a pipeline to generate a large-scale catalogue of

smORFs in microbes. The library has the potential to serve as a resource for the microbiome community.

General comments:

The manuscript provides a potentially useful resource of putative microbial smORFs. The resource is fairly well characterized and investigated. However, the manuscript is not easy to follow in terms of how it presents both its methods and results. Many method descriptions are in the Results section or figure captions, while many results are reported in the Methods section. This makes for a hard read of the manuscript. Furthermore, very few criteria used for building this catalogue are justified. The majority of criteria are arbitrary thresholds selected by the authors. Finally, little is done to convince the reader about the validity of the catalogue and its potential use. I provide more specific comments below.

Author response: We thank the reviewer for their appreciation of the usefulness of the resource, while also acknowledging their concerns, which we address below.

Major comments:

A large number of results are provided in the Methods section. These included, but are not limited to, results provided at line 257, 262, 269, 281, 286, 289, 299, 308, and 313. On the other hand, methods are often described better or repeated in figure captions. A significant reorganization of the text is needed to ease reading.

Author response: Thank you for pointing out the unclear parts of our text. We have reorganized our text structure including Results, Methods, and Figure captions. We removed and simplified the additional results from the Methods section which are already in the figures or the Results section to make it more concise and clearer to follow.

Changes made (main points, summarized): We moved the detailed numbers of each step of our pipeline for constructing our catalogue from the Methods section to the caption of figures. We removed the detailed numbers from the Methods section when they are shown in the figures. Specifically, in Methods, we removed the number of rescued singletons and non-singletons which are already shown in Figure 1a, and the number of smORFs that passed each quality test which is already shown in Supplementary Fig. 3 and Results section. We moved the detailed results of the significance validation of clusters from the Methods section to the Caption of Supplementary Fig. 1a-b.

Moreover, criteria and thresholds for metatranscriptomics, ribo-seq, and metaproteomics smORF confirmation are quite arbitrary. Instead of reporting a single number of confirmed smORFs at a selected threshold. Plots showing the number of smORFs passing at varying thresholds would provide a better grasp of the dataset, and would help providing a reasoning behind the choices of the different thresholds.

Author response and Changes made: Unfortunately, there are no well-validated standards in this field. In the case of interpreting the outputs of RNAcode and metaproteomics, we

used thresholds previously used in the literature (Sberro et al.; 2019; <https://doi.org/10.1016/j.cell.2019.07.016>; Ma et al. 2022; <https://www.nature.com/articles/s41587-022-01226-0>), but we acknowledge that this is not a consensus in the field and, furthermore, no analogous examples exist for thresholds applicable to transcriptional or translational data.

Therefore, we thank the reviewer for their suggestion, which we implemented: we now add a new **Supplementary Fig. 4** (reproduced below) to show how different thresholds lead to different numbers of high-quality predictions. We also make all this information available on the updated website for both download and interactive exploration (**Reviewer Figure 1**). While we kept our previous thresholds as defaults, users can now choose different combinations of parameters for their queries.

As part of this effort, since we had not saved all the intermediate results, we needed to rerun some of the quality checking. In addition, now we directly screened for proteomic coverage value without retaining one decimal place to make the results more accurate. Given that the results are not completely deterministic, this led to some very minor updates in the resulting high-quality counts.

Supplementary Fig. 4 (reproduced here for convenience). Effect of different thresholds on quality control (a) The number of smORFs with high coding potential as estimated by RNAcode, using different P-value thresholds. (b) The number of smORFs with transcriptional evidence, using different thresholds for the minimal number of samples required for detection. (c) The number of smORFs with translational evidence, using different thresholds for the minimal number of samples required for detection. (d) The number of detected smORFs in metaproteomics data, using different thresholds for the required k-mer coverage of each smORF-encoded small protein (Methods).

Reviewer Figure 1. Screenshots of the updated website showing the quality information. (a) Quality filtering interface when browsing/searching, (b) Results for a single cluster, showing details.

The smORFs family construction requires a more detailed explanation. What are the sequences that are clustered? Is it that any sequence that have at least one other sequence with which it has a 90% identity and 90% coverage is used as input for the clustering analysis? You could have three sequences named, A, B and C, with A and B having a 90% identify, A and C also having a 90% identity, but B and C not having this level of identity. Would these be all grouped together?

Author response: We used the standard pipeline Linclust (Steinegger & Söding, 2018; <https://doi.org/10.1038/s41467-018-04964-5>), which uses a greedy approach, whereby sequences are compared to candidate representatives. Thus, in the reviewer's example, if A was chosen as a potential representative, it would indeed be chosen as a representative for both B and C, even if B and C do not share this level of identity. Due to the very large size of

the input databases, such an approach is necessary to keep the computational costs reasonable.

Changes made: We now describe the Linclust algorithm as a heuristic single-linkage in Line 292 (novel text in bold) of Methods to make the clustering process easier to understand: “**Then** we hierarchically clustered the non-singletons at 90% amino acid identity and 90% coverage **using Linclust** with the following parameters: -c 0.9, --min-seq-id 0.9. **Linclust is a single-linkage approach, whereby sequences are clustered together if they share a common representative with candidate representatives being chosen heuristically.**”

The procedure used to evaluate the significance of the clusters appears convoluted, under sampled and arbitrary. Why not using a simple bootstrapping approach to evaluate the robustness of the clusters? This is a lot more standard and typical for such analyses.

Author response: Unfortunately, even after consulting with colleagues, we are not sure what simple bootstrapping procedure the reviewer may have had in mind. Perhaps we had not explained the purpose of the cluster evaluations sufficiently:

Our major concern was that, even though we are using a well-established pipeline for clustering (Linclust by Steinegger & Söding, 2018, see above), this pipeline was developed and benchmarked for canonical-length proteins. Therefore, we feared that some results (e.g., the fact that we observe a relatively large fraction of singleton clusters) could be due to us using it inappropriately (namely on small sequences). We wanted to estimate the rate of false negatives (i.e., sequences that were marked as singleton even though they should have been clustered with another one) and false positives (sequences that are members of a cluster even though they do not belong there). It is impossible to perform an exhaustive search for the whole catalogue, so we applied an exhaustive search method to a small, randomly chosen, sample to estimate these false negative/false positive rates.

Changes made: The corresponding section in the Methods (Line 301, novel text in bold) now reads: “Of these clusters, 47.5% contain a single sequence (singleton clusters). To rule out the possibility that this was due to the fact that Linclust is a heuristic method that is not specifically designed for short sequences, we **estimated the rate of false negatives (i.e., sequences that were marked as singleton even though they should have been clustered with another one).** We aligned a randomly selected 1,000 singleton clusters against the representative sequences of non-singleton clusters (i.e., those containing ≥ 2 sequences) using SWIPE with the following parameters: -a 18 -m '8 std qcovs' -p 1. The alignment threshold was E-value $< 10^{-5}$, identity $\geq 90\%$, and coverage $\geq 90\%$ (Supplementary Fig. 1a).

In addition, to estimate the rate of false positive clusterings (sequences that were assigned to a cluster even though they do not share the required identity with the cluster representative), 1,000 sequences were randomly selected and aligned against the representative sequences of their clusters using SWIPE with the following parameters: -a 18 -m '8 std qcovs' -p 1. The alignment threshold was E-value $< 10^{-5}$, identity $\geq 90\%$, and coverage $\geq 90\%$ (Supplementary Fig. 1b).”

Also how are the representative sequences of the clusters determined? This is not clearly described in the methods.

Author response and changes made: We used the Linclust pipeline, which implements a heuristic approach to choose representatives (Steinegger & Söding, 2018; <https://doi.org/10.1038/s41467-018-04964-5>) as we now mention on Line 292 (see above). A detailed explanation of the Linclust heuristic can be found in its manuscript.

How does the proposed catalogue compare to that of OpenProt and the method they used for their database construction (Sébastien Leblanc, Ferial Yala, Nicolas Provencher, Jean-François Lucier, Maxime Levesque, Xavier Lapointe, Jean-François Jacques, Isabelle Fournier, Michel Salzet, Aïda Ouangraoua, Michelle S Scott, François-Michel Boisvert, Marie A Brunet, Xavier Roucou, OpenProt 2.0 builds a path to the functional characterization of alternative proteins, *Nucleic Acids Research*, Volume 52, Issue D1, 5 January 2024, Pages D522–D528, <https://doi.org/10.1093/nar/gkad1050>). OpenProt did not cover bacteria so far and they do not have the same objectives, but some of them are overlapping. Would their approach be applicable here?

Author response: As the reviewer pointed out, the OpenProt2.0 database contains sequences from eukaryotic model organisms with expressed and translated evidence and comprehensive functional annotation. Due to thorough research on these eukaryotic model organisms, comprehensive transcriptome data is available in NCBI RefSeq and Ensembl. However, for prokaryotes, many species are uncultured and even lack complete genomes. Transcriptomics, proteomics, and other non-genomic data types are only sporadically available for most of the organisms/communities that we study here. Therefore, we use high-quality microbial genomes and assembled metagenomes instead of transcriptomes to construct the catalogue of microbial smORFs.

Nevertheless, we also checked the expressed and translated evidence of smORFs in our catalogue using existing paired metatranscriptomic data when available. In addition, we annotated conserved domains for small proteins using the Conserved domain database (CDD). As the reviewer wrote, we do share some objectives with the OpenProt2.0 database, and will take some inspiration for future work (e.g., exploring linear motifs, intrinsic disorder, or structure prediction), but the settings are very different.

Changes made: We compared our catalogue with OpenProt2.0 to understand how much overlap there is between prokaryotic and eukaryotic small proteins. We added the comparison results in Supplementary Fig. 5c and mentioned in Line 233 of Discussion, as below “On the other hand, it encompasses most of the known small proteins in either the RefSeq database or in families discovered recently (NMPfamsDB and FesNov families). When comparing with small protein databases that focus on eukaryotic organisms, such as smProt2, OpenProt2.0, and sORF.org, the overlap is minimal (Supplementary Fig. 5c).”

Supplementary Fig. 5 (reproduced here for convenience, compared to the previous version, panel c was added; note that this was previously Sup. Fig. 4). Comparison of reference small protein datasets (a) Shown is the fraction of smORFs from high-quality predictions that are homologous to reference small protein datasets. (b) The comparison of the proportions of smORFs from human or non-human habitats between homologs or non-homologs to small protein clusters and conserved families from the Sberro human microbiome dataset. (c) Shown is the fraction of GMSC smORFs that are homologous to NMPfamsDB, FesNov families, smProt2, OpenProt2.0, and sORF.org.

The data does not back-up the following conclusion stated by the authors: "Archaea have more transmembrane or secreted small proteins than bacteria". First, no evidence is provided that these specific smORFs are translated into proteins. Second, archaea have way less data points and I would assume that if one would remove a couple of the highest points that look more like outliers than anything else, the result would not be significant anymore. This result appears to be an artefact of the methods used to identify smORFs and transmembrane domains more than anything else. The conclusion of this entire section should be removed or rewritten.

Author response: We agree with the reviewer that the previous version had overstated what can be shown from the data. While we tested whether it was a statistical artifact by removing outliers and, from that perspective, the result is solid (see **Reviewer Figure 2**), given the nature of the project, we are not able to rule out the possibility that our methodology (used to identify smORFs or classify them as coding for transmembrane proteins) has a bias that affects these two domains differently. Therefore, we have removed that conclusion and only report that tools return different predictions. We would also like to point out that in the Discussion, we had already considered that the tools we used for predicting whether a small protein is transmembrane/secreted were not optimized or benchmarked for this setting (Line 259) and we now also consider that we rely on the assumption that error rates are similar between archaea and bacteria.

Reviewer Figure 2. (a, same as Fig. 4a) Boxplot showing the fraction of transmembrane or secreted small proteins in bacteria and archaea, p-value obtained from Mann-Whitney test. **(b)** Boxplot with outliers removed showing the fraction of transmembrane or secreted small proteins in bacteria and archaea, p-value obtained from Mann-Whitney test. (Outliers are defined by Tukey's fences based on interquartile range).

Changes made: We have renamed both the subsection and Figure 4 from “Archaea have more transmembrane or secreted small proteins than bacteria” to “Differences in functional prediction for archaeal and bacterial small proteins.” We have rephased the Results section to now read (Line 175, novel text in bold): “15.3% of the families are predicted to be transmembrane (using TMHMM-2.0) or secreted (using SignalP-5.0), **with archaeal families being predicted at a higher rate than bacterial ones to be transmembrane or secreted ($P_{mann} \leq 0.0103$, Fig. 4b).**” In Line 259 of the Discussion section, we now added the following consideration: “In particular, when we compared results between bacteria and archaea, we implicitly assumed that the methods have similar error rates in these two domains, but this may not be the case.”

To provide further insights into the validity of the catalogue, I would have expected that sequence conservation would have been directly investigated.

One would assume that high-quality predictions are more likely to be functional than low-quality ones. Hence, they should be more likely to be evolutionarily conserved. Is it the case that nucleotides part of these high-quality predictions are more conserved than those that are of lower quality. A fold-enrichment could be provided to yield such an assessment.

Author response: We checked for this and found only a very minimal effect. While, as predicted by the reviewer, there is a higher level of conservation at the nucleotide level, the differences are small (see **Reviewer Figure 3a**). We also attempted to estimate the ω (dN/dS) ratio and found only a negligible difference (see **Reviewer Figure 3b**). While the direct comparison of ω values is statistically significant, we note that the effect size is small and alternative specifications (e.g., the fraction of families with $\omega < 1.0$, which indicates negative selection) are not statistically significant, with 86.69% for high-quality families and 86.47% for others.

Reviewer Figure 3. (a) Sample of 1,000 HQ and non-HQ families. For each family, a random pair of elements was compared as representative (p-value estimated by Mann-Whitney two-sided test). **(b)** ω (dN/dS) as estimated by codeml program of the PAML (Yang, Z., 2007; <https://doi.org/10.1093/bioinformatics/13.5.555>) for 1,000 HQ and non-HQ families (p-value estimated by Mann-Whitney two-sided test).

In order to provide some insights into the potential applications and discovery potential of the catalogue, it would be interesting to see how these novel smORFs can help identify more peptides and proteins in metaproteomics studies. Most mass spectrometry-based metaproteomics studies will identify proteins using a technique called sequence database search. Providing a set of smORFs not typically included in such sequence database searches could help reveal new proteins never identified in metaproteomics datasets in the past.

Author response: We agree with the reviewer that this is an important avenue of exploration. Working with Benoît Kunath and Paul Wilmes (University of Luxembourg), we

attempted to re-analyze a previously collected metaproteomics dataset for a human gut microbiome sample for which there was a corresponding metagenome available. We used the usual database augmented with the smORFs matched GMSC that we predicted from the contigs of the corresponding metagenome. While we could find 462 smORFs identified by metaproteomics, only 13 unique smORFs were found. We (and others) have ongoing efforts to improve the small preparation and bioinformatics to specifically target small proteins, but this is beyond the scope of the current manuscript.

(Data shared courtesy of Benoît Kunath and Paul Wilmes, University of Luxembourg).

No ReadMe are provided with the code, making its evaluation extremely difficult.

Author response and changes made: Now, we provided the complete ReadMe files to facilitate tracking and reproducing analysis. In the main ReadMe file, we provided all the dependency tools and databases for processing analysis. In the General_Scripts folder, we provided ReadMe files containing the names of codes, the description of codes, the inputs of codes, and the outputs of codes.

Minor comments:

Why were 10,000 randomly selected prokaryotic proteins queried using RPS-BLAST?

Author response: Indeed, this step was described in the Methods, but not linked to its context. As we now clarified, we wanted to estimate the fraction of canonical-length proteins that are assigned CDD domains to put the corresponding results on small proteins in context. The result is mentioned in Line 117 “Only 6.1% of small protein families containing 86,694,259 smORFs (8.98%) were assigned CDD domains, compared to 35.2% of canonical-length proteins (greater than 100 amino acids)”.

Changes made: To make the link to the corresponding analysis clearer, the relevant sentence (Line 408) in the Methods now reads “In order to establish a comparison baseline, we additionally randomly selected 10,000 prokaryotic proteins from the global microbial gene catalogue v1.0 and searched them against the Conserved domain database by RPS-BLAST”

Versions used should be provided for Python, Pandas, NumPy, and SciPy.

Author response and changes made: We thank the reviewer for pointing out these. Now we have updated the version of these tools in Line 480 of Methods. “Statistical analyses were carried out in Python 3.8.5, using Pandas 1.1.3, NumPy 1.24.4, and SciPy 1.10.1.”

Reviewer #2 (Remarks on code availability):

The code was not thoroughly reviewed due to a lack of instructions on how to execute it.

Author response and changes made: Now, we have reorganized the codes and pre-calculated data along with the complete ReadMe files to facilitate tracking and reproducing analysis. The General_Scripts folder contains scripts to generate the GMSC resource. The Manuscript_Analysis folder contains pre-computed files and scripts to run the analysis and generate the main figures and supplementary figures included in the GMSC manuscript. In the main ReadMe file, we provided all the dependency tools and databases for processing analysis. In the General_Scripts folder, we provided ReadMe files containing the names of codes, the description of codes, the inputs of codes, and the outputs of codes.

REVIEWERS' COMMENTS

Reviewer #1 (Remarks to the Author):

Well done to the authors for careful responses to reviewers, and including new work such as comparisons to other databases.

I only have one minor note remaining:

I find e.g. Supp. Figure 5 difficult to view, because of partial red/green colour-blindness, which is very common. Perhaps other colours could be used if the data & code for this Figure is readily available. It seems less of an issue for other figures (perhaps because the orange bars are larger) and because consistency is probably wanted across figures this is up to the authors' discretion.

Reviewer #1 (Remarks on code availability):

Code is well organised and described. The use of Jupyter notebooks aids in the interpretability.

Reviewer #2 (Remarks to the Author):

The authors answered my concerns.

Reviewer #2 (Remarks on code availability):

A Readme is now provided.

Point-by point response

Reviewer #1 (Remarks to the Author):

Well done to the authors for careful responses to reviewers, and including new work such as comparisons to other databases.

I only have one minor note remaining:

I find e.g. Supp. Figure 5 difficult to view, because of partial red/green colour-blindness, which is very common. Perhaps other colours could be used if the data & code for this Figure is readily available. It seems less of an issue for other figures (perhaps because the orange bars are larger) and because consistency is probably wanted across figures this is up to the authors' discretion

Author response and changes made: We thank the reviewer for pointing out the color used in the figures. We originally intended to choose a colour-blind friendly palette from Colorbrewer. However, while generating figures, some of the colors in the figures showed slight color shifts. Now we have checked and fixed the colors of all the figures using the correct palette.

Reviewer #1 (Remarks on code availability):

Code is well organised and described. The use of Jupyter notebooks aids in the interpretability.

Author response: We thank the reviewer for the positive feedback on our codes.

Reviewer #2 (Remarks to the Author):

The authors answered my concerns.

Reviewer #2 (Remarks on code availability):

A Readme is now provided.

Author response: We thank the reviewer for the positive comments and for recommending accepting our manuscript.